

# Understanding and mitigating the impact of data gaps on offshore wind resource estimates

Julia Gottschall[a] and Martin Dörenkämper[b]

[a]Fraunhofer IWES, Am Seedeich 45, 27572 Bremerhaven, Germany
[b]Fraunhofer IWES, Küpkersweg 70, 26129 Oldenburg, Germany

**Correspondence:** julia.gottschall@iwes.fraunhofer.de

**Abstract.** Like almost all measurement datasets, wind energy siting data are subject to data gaps that can for instance originate from a failure of the measurement devices or data loggers. This is in particular true for offshore wind energy sites where the harsh climate can restrict the accessibility of the measurement platform, which can also lead to much longer gaps than onshore. In this study, we investigate the impact of data gaps and its mitigation by correlation and filling with mesoscale model data. Investigations are performed for three offshore sites in Europe, considering two years of parallel measurement data at the sites, and based on typical wind energy siting statistics. We find a mitigation of the data gaps' impact by a factor of ten on mean wind speed, direction and Weibull scale parameter, and a factor of three on Weibull shape parameter. With increasing gap length, the gaps' impact increases linearly for the overall measurement period while this behaviour is more complex when investigated in terms of seasons. This considerable reduction of the impact of the gaps found for the statistics of the measurement time series almost vanishes when considering long-term corrected data, for which we refer to 30 years of reanalysis data.

## 1 Introduction

A wind resource assessment is performed at the beginning of every wind energy project. The wind resource is estimated for the site that is pre-selected with respect to the expected lifetime of the project, i.e. for the 20-30 years in the future during which the wind turbines will be operated at the site (Rohrig et al., 2019). Based on this estimate an expected energy yield is derived which serves as a basis for any economic considerations of the project. Consequently, uncertainties and a possible bias in the wind resource estimate propagate up to the financing of a wind project, and to reduce them is of high interest and relevance.

A wind resource assessment is typically based on a short-term measurement on site, that is conducted several years prior to the installation of the wind farm and has a duration in the order of a year (MEASNET, 2016; FGW e.V., 2017). The campaign duration is in most cases a compromise between informative value – defined by the representativeness of the measurements for the lifetime of the wind farm, i.e. those 20-30 years in the future – and the costs of the measurement campaign. In a later step, short-term measurements are long-term extrapolated making use of a reference data set that is either a longer multi-year measurement in the surrounding area of the site or data from a reanalysis, sometimes downscaled with the use of a mesoscale model, with a resolution of several (tens of) kilometers around the site (Carta et al., 2013). In case of complex terrain or



differing measurement heights, horizontal and vertical interpolation is done using numerical computational fluid dynamics (CFD) and/or simplified engineering models (Rohrig et al., 2019).

Almost all measured time series have data gaps due to failures of the sensors themselves, a data logger or the power supply, or due to adverse conditions as e.g. a low aerosol concentration or unwanted fixed echoes for remote sensing devices (MEASNET,
2016). In case of offshore measurements, data gaps often further increase due to limited accessibility to the measurement installation in particular in high wind and wave conditions that may typically last for several weeks or even prevent access for a whole season. Additionally, many offshore wind measurements are due to their high costs not fully redundant, which is particularly the case for many floating lidar applications that prevail more and more in the offshore wind industry as a most cost-efficient alternative to fixed offshore meteorological (met) masts (Gottschall et al., 2017).

Up to a certain threshold of amount of lengths of data gaps, the long-term extrapolation, which in the standard procedures involves some correlation of measured and reference data for the overlapping period, is often applied to the not fully continuous time series. MEASNET (2016), for instance, considers a measurement as incomplete only when the availability of filtered data is less than 90 %. As an alternative, the time series can be "filled" before the application of the correlation analysis. Gap-filling procedures typically use reanalysis data, e.g. from MERRA2 (Donlon et al., 2012) or ERA5 (Hersbach and Dick, 2016), often
downscaled with a mesoscale model as e.g. WRF (Skamarock et al., 2019). Such a "gap filling" is, in particular, applied when the gap corresponds to a substantial discontinuity in a measurement time series of several days, weeks or even months, not just a few data points that can be filled by statistical approaches or even interpolation only.

Gap filling is a task that is not specific to the wind resource assessment application but can be of relevance for any measured time series or collected dataset where data gaps may significantly impact the outcome of the following data analysis. In the
most general context, procedures to compensate missing values in a dataset are referred to as imputation. There are a number of different imputation procedures that have in common that missing data is not simply ignored but instead replaced by plausible values. Specific gap filling procedures for meteorological time series are e.g. discussed in Körner et al. (2018), Pappas et al. (2014) and the references herein – these include

- linear interpolation from adjacent time steps (particularly for cases where only a few data points are missing),

- autoregressive models (for longer periods of missing data and without adjacent sites as possible predictors),

- different methods of spatial interpolation (in case adjacent sites are available),

- data-driven methods like nearest-neighbour approaches, linear or multiple linear regression, look-up tables or artificial neural networks (Körner et al., 2018).

For the wind resource assessment application, linear regression methods are of particular interest since they are often already
used for the long-term extrapolation in the context of measure-correlate-predict (MCP) approaches (MEASNET, 2016). Often further dimensions are introduced by considering separate wind direction sectors or wind speed bins. Generally, MCP methods are not limited to linear regressions (see Carta et al., 2013, for a broader overview), however, in practice they are most often





implemented in this way. This is why we concentrate on this type of procedure – both for data gap filling and long-term extrapolation – in this contribution.

The overall scope of the study is as follows: before discussing a selected specific data gap filling approach, we investigate how data gaps impact the standard wind resource estimates by deriving and evaluating bias and uncertainty measures for wind time series with artificial gaps of varying length and seasonal period of occurrence. We repeat this analysis for the time series where the gaps are filled and, with this, study to which extent the impact of the gaps can be mitigated. The study is applied to the statistics of the short-term dataset, defined by the period of the measurements, as well as to the final long-term estimate, since both sets of results are relevant in the wind energy context. By deriving and comparing conclusions for three different offshore sites — in the German Bight, the Dutch North Sea and in the Baltic Sea — we also address the impact of the site and possible dependencies.

The article is structured as follows: in Section 2 we describe the data basis and in Section 3 the methods for this study. Section 4 presents the results for the impact of ignored and filled gaps on the short-term and long-term wind statistics. In Section 5 we discuss our findings with the particular implications for future resource assessment studies. And with Section 6 we conclude our contribution.

## 2   Data basis

The data basis consists of measurement data from meteorological (met) masts over a measuring time which is characterized by a high availability on the basis of which the influence of measurement gaps and their filling is investigated (subsection 2.1), as well as numerical data used for filling the gaps in the measurement data and the subsequent long-term extrapolation (subsection 2.2).

### 2.1   Sites and measurement data

The analyses in this study are done independently for three different offshore met masts representing different typical sites for offshore wind energy utilisation in Europe. Two masts are located in the North Sea (FINO3 and Ijmuiden) both about 50 km offshore from the next coastline with large wind direction sectors where the next coastline is several hundreds of kilometres upstream. The third mast (FINO2) is located in the central Southern Baltic Sea and surrounded by land within 50 km or less except for a small wind direction sector. The sites were chosen to represent typical European offshore wind exploration areas with different distances to the coasts and varying atmospheric stability (see e.g. Dörenkämper, 2015; Kalverla et al., 2019).

In Figure 1 the positions of the three met masts are given. The frames mark the area of the innermost domains of the mesoscale data used for the gap filling (see Section 2.2). The data of all three met masts are freely available for scientific purposes. All of the masts are equipped with cup or sonic anemometers and wind vanes up to a height of 100 m above the sea surface. For our study we consider the 10-min averages of horizontal wind speed and direction data provided by these sensors. A measurement height close to 90 m was chosen at all three masts (see description below for more details), representing a typical hub height of offshore wind turbines. The 24-month period from 01-07-2012 to 30-06-2014 was selected based on the



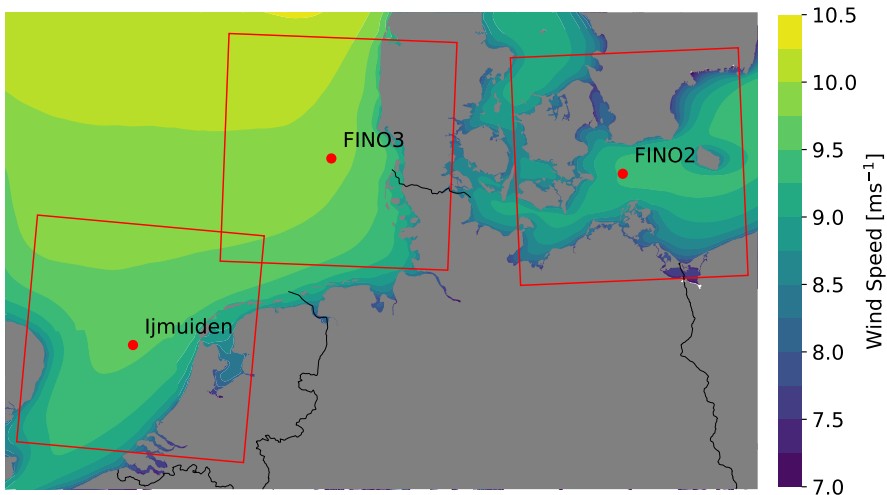

**Figure 1.** Position of the three sites (met masts) investigated in the framework of this study. The red boxes mark the sizes of the innermost domains used for the mesoscale modelling (cf. 2.2). The background wind field represents the 30 year mean wind speed (1989-2018) at 100 m height above the sea surface taken from the New European Wind Atlas (Hahmann et al., 2020; Dörenkämper et al., 2020).

combined data availability for all three masts and other constraints such as limiting disturbance of wakes of nearby wind farms that were erected afterwards in the vicinity to the masts some years after the commission of the respective masts.

As the aim of this study is to investigate the impact of gaps on offshore wind energy relevant wind statistics, a reference time series with a low amount of missing data was needed. Thus, besides the selection of the two years period with a low amount of

gaps, further gaps were filled with measurement data from lower altitudes. To consider the wind speed dependence with height, a speed-up factor sup is defined according to $\mathrm{sup} = \mathrm{WS90}_{\mathrm{mean}}/\mathrm{WS[X]}_{\mathrm{mean}}$, where $\mathrm{WS90}_{\mathrm{mean}}$ is the average wind speed at the measurement altitude of the mast closest to 90 m and $\mathrm{WS[X]}_{\mathrm{mean}}$ the average wind speed of the measurement at a lower height, and applied to the wind speed measurement of the lower altitude. In case of gaps, the wind direction measurements were filled by measurements at lower heights as well but without using any scaling or offset correction. Data gaps in the mast

measurements filled by applying this pre-processing are shown in Figure 2. A short description of the three met masts with references to more detailed information is given in the following:

– The **Ijmuiden** met mast[a] is located about 85 km west of Den Helder in the Dutch part of the North Sea. The met mast was in operation between November 2011 and March 2016 and was decommissioned afterwards. The mast was used in several wind energy research studies (e.g. Baas et al., 2016; Kalverla et al., 2019). It provides measurements at several

heights and is described in more detail in Poveda et al. (2015). For the analysis in this study, the mast corrected wind speed measurement (cup anemometers) located at 92 m height was used together with the wind direction measurement

---

[a]https://www.windopzee.net/en/meteomast-ijmuiden-mmij/index.html





at a height of 87 m. The availability of the wind speed measurement data at these heights is 99.5 % prior and 99.7 % after the filling of gaps by lower measurement heights (see procedure above).

- The **FINO2** met mast[b] is located in the central Southern Baltic Sea close to the border triangle of Denmark, Germany and Sweden in the German part of the Baltic Sea. In contrast to the North Sea sites, the FINO2 measurements are affected

by the surrounding lands with distances of less than 50 km for the majority of wind direction sectors. Only a narrow northeasterly sector is dominated by a long marine fetch. FINO2 is in operation since August 2007 and the data were studied in several wind energy related studies (e.g. Gryning et al., 2014; Dörenkämper, 2015). FINO2 provides wind measurements at various heights between 32 m and 102 m above sea level, technically described in (FINO2, 2007). In this study mainly the wind speed measurements from the cup anemometers at 92 m height were used in combination with

the wind direction measurement (vane) at the same height on the boom of the opposite side of the mast. The availability of the wind speed time series was 86.4 % prior and 95.5 % after the application of the gap filling from lower heights (see procedure above).

- **FINO3** is a met mast[c] located in the northern part of the German Bight about 80 km north west of the island of Sylt. Thus, the impact of upstream coastlines is very limited and a pure offshore climate is found in particular for the main wind

direction sectors (south to northwest). FINO3 is in operation since September 2009 and provides wind measurements each 10 m between 32 m and 102 m above sea level as described in (FINO3, 2012). The FINO3 wind measurements were part of several wind energy studies (Peña et al., 2015; Gryning et al., 2016). This study analyses the wind speed (cup) and wind direction (vane) data from the measurements at 92 m respectively 101 m above mean sea level. These wind speed measurement data have an availability of 98.4 % prior and 98.9 % after the application of gap filling from measurements

at lower heights (see procedure above). A detailed overview of the measurements of the three FINO masts, their device types, accuracy and boom orientations is given in the Appendix of Leiding et al. (2012).

Wind speed and wind direction distributions for the three datasets of measurements are shown in Figure 3. Note that the measurement heights slightly differ for the three sites – as described above, wind speed measurements are recorded at 92 on all three masts, while wind directions are recorded at 87 m (Ijmuiden), 92 m (FINO2) and 101 m (FINO3), respectively.

Derived wind statistics are summarized in Table 1. Here and for the following analysis we consider the parameters *mean wind direction*, *mean wind speed*, and the parameters $k$ (Weibull shape parameter) and $A$ (Weibull scale parameter) that are obtained from fitting a Weibull distribution function to the wind speed distributions considering the complete wind speed range. All three masts represent typical mid-latitude offshore wind climates. A shift from southwesterly to more westerly winds is found while moving from west to east, being in-line with the typical track of cyclones when moving across central Europe (van Bebber,

30   1891).

---

[b]https://www.fino2.de/en/
[c]https://www.fino3.de/



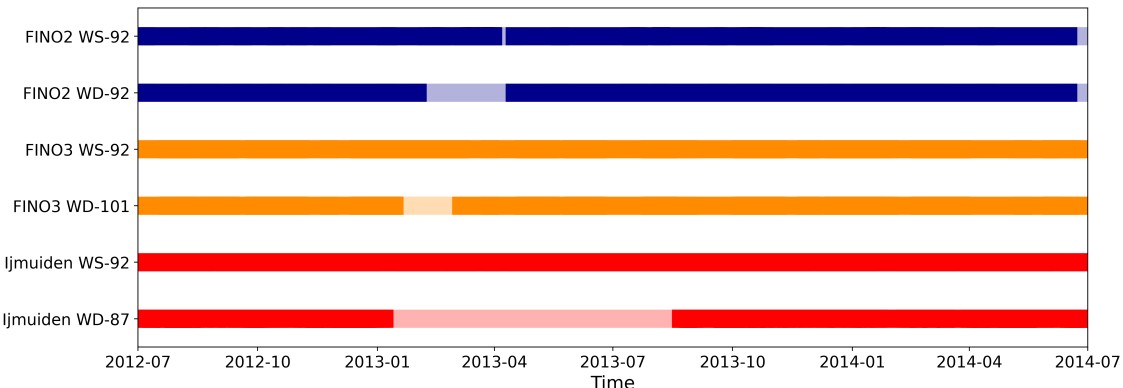

**Figure 2.** Data availability at the three masts after filling in the pre-processing step. The light colors indicate values that were filled by measurements from lower heights as described above.

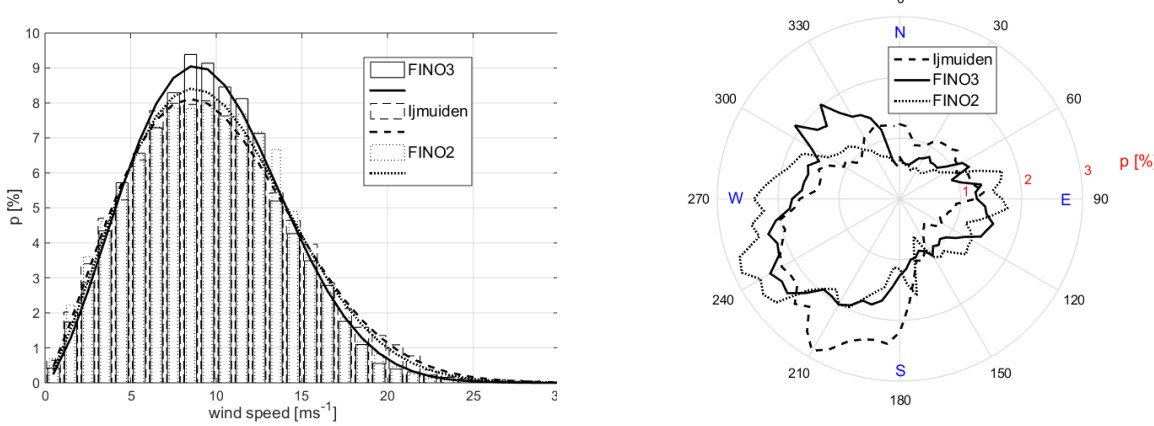

**Figure 3.** Wind speed (left) and wind direction (right) distributions for the three 24-month datasets from the FINO3, Ijmuiden and FINO2 offshore met mast sites. Derived statistics are summarized in Table 1.

## 2.2 Numerical data for gap filling and long-term extrapolation

The procedures applied for this study make use of regional mesoscale modelling data that are used for the gap filling, as well as long-term reanalysis data that are applied for long-term referencing of the wind measurements. These data sources are described separately below.

### 2.2.1 Reanalysis data

For the long-term extrapolation, the data from the ERA5 reanalysis (Hersbach et al., 2020) were used. ERA5 is the most recent generation of reanalysis data issued by the European Centre for Medium-Range Weather Forecasts (ECMWF) since 2017. For



**Table 1.** Derived statistics for wind direction and wind speed distributions for the three datasets of site measurements.

| Site | mean wind direction [deg] | mean wind speed [ms$^{-1}$] | Weibull parameter $k$ [-] | Weibull parameter $A$ [ms$^{-1}$] |
|---|---|---|---|---|
| FINO3 | 243.6 | 9.60 | 2.44 | 10.92 |
| Ijmuiden | 233.4 | 9.88 | 2.19 | 11.24 |
| FINO2 | 228.3 | 9.59 | 2.28 | 11.18 |

wind energy applications it was shown to outperform other reanalyses (e.g. Olauson, 2018; Thøgersen et al., 2017). ERA5 provides reanalyses on all important atmospheric and oceanographic parameters in an hourly resolution in time and 0.25 ° ($\approx 30$ km for the atmospheric parameters, others differ) in longitudinal and lateral direction globally. Currently the period of 1979-ongoing is publicly available with a lag of a few days in time. For the long-term referencing in this study, the wind

speed (lateral and longitudinal components, $u$ and $v$ at 100 m) from the so-called surface level data of the ERA5 dataset were selected for the period 1983-2014 to cover a climatic period of 30 years. Most recently, mesoscale model datasets with a higher resolution in the order of a few kilometres were made available for longer periods (up to 30 years) and sometimes used for long-term referencing. However, as the industry at least partly still relies on classical lower resolution reanalyses, we have applied this approach for our study. In addition, due to its comparatively high resolution ERA5 does not show major differences in

the offshore wind speed climate statistics several tens of kilometres away from the coastal discontinuity (Dörenkämper et al., 2020).

### 2.2.2 Mesoscale modelling and data

The mesoscale model data in this study are used for filling the gaps that are artificially cut into the time series. In principle any mesoscale model data could be used such as those from the publicly available New European Wind Atlas (NEWA) (Hahmann

et al., 2020; Dörenkämper et al., 2020) or commercial products. However, these data are often not optimized for offshore wind energy applications or only available in lower resolution in time (e.g. 30 min instead of the desired 10 min data). Consequently, simulations were performed separately for this study applying a setup that was optimized for offshore wind applications (Dörenkämper et al., 2015, 2017; Gottschall et al., 2018) and capable in resolving the most important flow features in offshore development regions.

The simulations were carried out using the Weather Research and Forecasting (WRF) model (Skamarock et al., 2019) in its version 4.0.1 which is generally well known and commonly used in the wind energy community (e.g. Dörenkämper et al., 2015; Hahmann et al., 2020). The mesoscale simulation setup was similar for every of the three sites, consisting of three domains with resolutions of 18 km, 6 km and 2 km and a domain size of 150 x 150 GP for each domain, centered around the site of interest (i.e. the respective offshore met mast). Figure 4 exemplarily shows the size of the three domains for the FINO3 site.

The sizes of the innermost domains (D3) for all three sites are given in Figure 1.



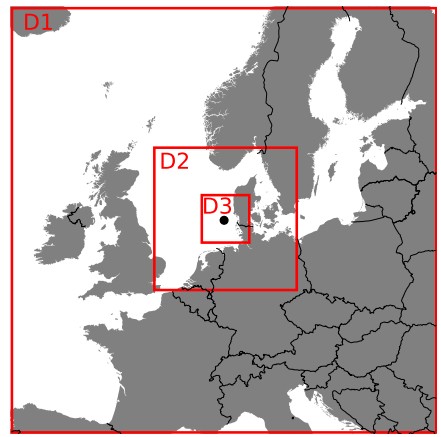

**Figure 4.** Mesoscale model domain distribution around the FINO3 site. The red boxes mark the extension of the computing domains.

Boundary conditions for the model were prescribed by the ERA5 dataset for the atmospheric variables (Hersbach and Dick, 2016; Hersbach et al., 2020) and the OSTIA dataset for the sea surface variables (Donlon et al., 2012). An instantaneous output of the mesoscale model on 10-min intervals was chosen, being consistent with the 10-min means of the met masts.

**Table 2.** Relevant parameters of the setup for the mesoscale simulations applied in this study. The references for the different schemes and models are summarized in WRF Users Page (2020).

| Parameter | Setting | Parameter | Setting |
|---|---|---|---|
| WRF model version | 4.0.1 | Planetary Boundary Layer scheme | MYNN level 2.5 |
| Land-Use data | Modis | Surface Layer scheme | MYNN |
| Atmospheric boundary conditions | ERA5 | Microphysics scheme | WRF Single-Moment 5-class |
| Sea surface conditions | OSTIA | Shortwave and longwave radiation | RRTMG |
| Horizontal resolution | 18 km, 6 km, 2 km | Nesting | one-way |
| Vertical resolution | 60 eta-level | Nudging | grid nudging above level 25 |
| Model output interval | 10 min | Land Surface Model | Unified Noah Land Surface Model |

Table 2 shows a summary of the most important model set-up parameters and thus boundary conditions and model physics
5   used in this study to drive the simulations. The output data on the WRF internal (Arakawa C, sigma terrain following) grid were converted to earth relative quantities using the post-processing script developed and verified in the framework of the NEWA project[d] (Dörenkämper et al., 2020). The data were interpolated to the exact measurement heights from the WRF levels, and virtual met masts were extracted at the grid point closest to the location of the met masts investigated in this study.

---

[d]https://github.com/newa-wind/Mesoscale/tree/master/postproc - last visit 13.07.2020





## 3 Applied procedures

The methods applied for our study are described in the following subsections and partly demonstrated on the basis of the FINO3 dataset.

### 3.1 Generation of artificial gaps

5 The first step of the analysis consists in generating artificial data gaps in the measured time series. This is demonstrated in Figure 5 in the two upper plots. A data gap is defined by its length (in days or e.g. number of 10-min intervals) and its start state. Figure 5 shows a gap of 30 d length starting on 30 September 2013 0000 UTC. For the results presented in Section 4 we have considered gap lengths between 6 and 90 d, and start dates running through the 2-year measurement duration in equal increments. The four for wind energy relevant statistical measures (mean wind speed and direction and Weibull shape and scale

10 parameters) already presented in 2.1 are derived for the incomplete time series in the same way as for the original ones but ignoring the data in the gap. The deviations in these measures represent the impact of the gap on the wind statistics. Note that for this study we have only considered single gaps of varying lengths. Multiple gaps are briefly discussed in Section 5.

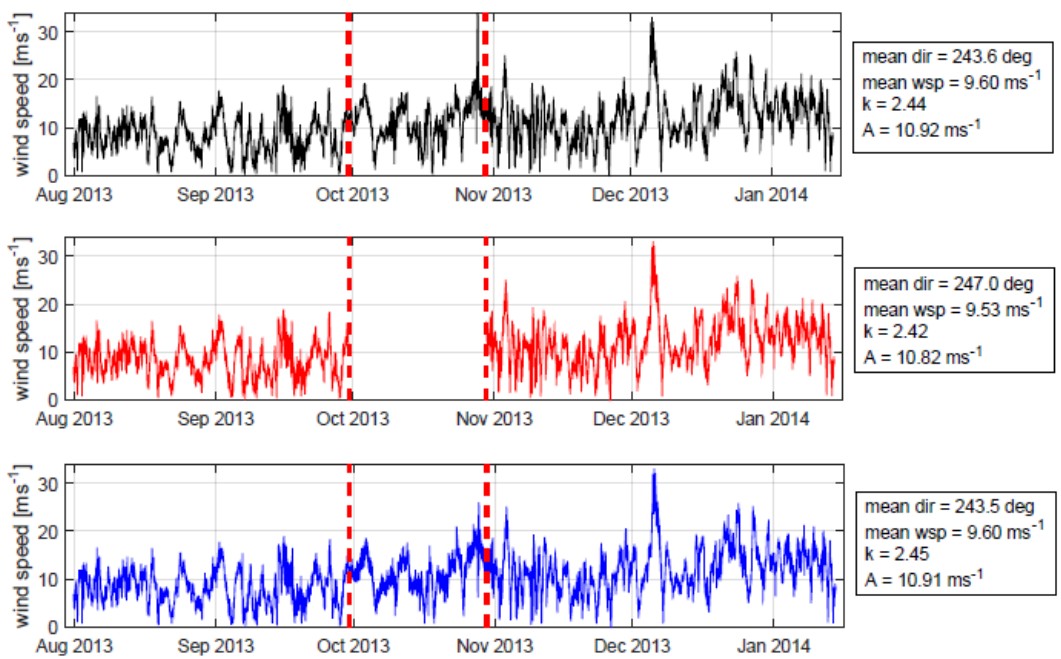

**Figure 5.** Generation and filling of artificial gaps – here demonstrated on the basis of the FINO3 wind speed time series and for a gap of 30 d length starting on 30 September 2013. Original time series (only a cutout is shown) in black, incomplete time series with generated gap in red, time series with filled gap (cf. procedure described in 3.2) in blue. Derived statistical measures (mean wind direction, mean wind speed, and $k$ and $A$ parameters of fitted Weibull wind distribution) are shown for the three time series on the right side.





## 3.2 Gap filling procedure

The artificial gaps are filled based on a measure-correlate-predict (MCP) procedure and with the WRF data introduced in 2.2 as input. The measured time series consist of the wind speed and direction data including the generated gap, respectively. These data are correlated with the numerical (WRF) data for the same period. That is, for the period of the gap no data is considered

for the correlation step. The correlation defines a correction that is implemented slightly differently for the wind speed and direction time series, respectively:

    – For the wind speed data, first the mean values per $0.5$-$\mathrm{ms}^{-1}$ bin are calculated and then a piece-wise linear fit is applied to these data points by fitting two linear functions to the wind speed ranges $[0,5)$ and $[5,20]\mathrm{ms}^{-1}$, respectively, and using the one or the other fit as a correction function depending if the reference wind speed is below or above the value of the

10        crossing of the two fits.

    – For the wind direction data, again first the mean deviation between measured and simulated wind direction per 10 deg bin is derived and then used directly as offset for the correction.

Note that the choice of this approach is more or less arbitrary but motivated by current practice for similar studies and applications. No further procedural steps as e.g. sector-wise corrections are considered. A corresponding higher-order approach was

considered unnecessary, as the sites under consideration are located far enough offshore to neglect directional effects.

    In addition to the correction factor – either resulting from the correction function or the bin-wise mean offset – a noise factor is derived as standard deviation of the data per bin, and combined with a white-noise process in the prediction step. For the prediction of the data in the gap period, numerical (WRF) data for this period are combined with the derived corrections. The resulting time series are inserted to the incomplete measurement time series. Figure 5 (bottom plot) shows the outcome of the

gap-filling procedure for the FINO3 time series used for demonstration.

    As already mentioned above, the procedure consisting of generating artificial gaps in the measured time series and the filling with the outlined MCP approach, is repeated for varying start dates of the gap that has the pre-defined length (in the example 30 d). This is demonstrated in Figure 6, where the four derived statistical measures are shown for an unchanged gap length but systematically varying start date for the incomplete and filled time series (in red and blue). The wind statistics for the original

time series are shown as a reference (as black lines). This example demonstrates that the impact of the generated gap varies quite drastically depending on when the gap starts, similar patterns are observed for all four considered statistical measures, and the gap-filling procedure is able to significantly reduce the impact in almost all cases. To quantify the observed variations in the wind statistics the corresponding root mean square error (RMSE) values are derived. For all four statistical measures these reduce when the gaps are filled: for mean wind direction from 3.1 to 0.3 deg, for mean wind speed from 0.07 to 0.01

$\mathrm{ms}^{-1}$, for the Weibull scale parameter $A$ from 0.09 to 0.01 $\mathrm{ms}^{-1}$, and for the Weibull shape parameter $k$ from 0.017 to 0.007.

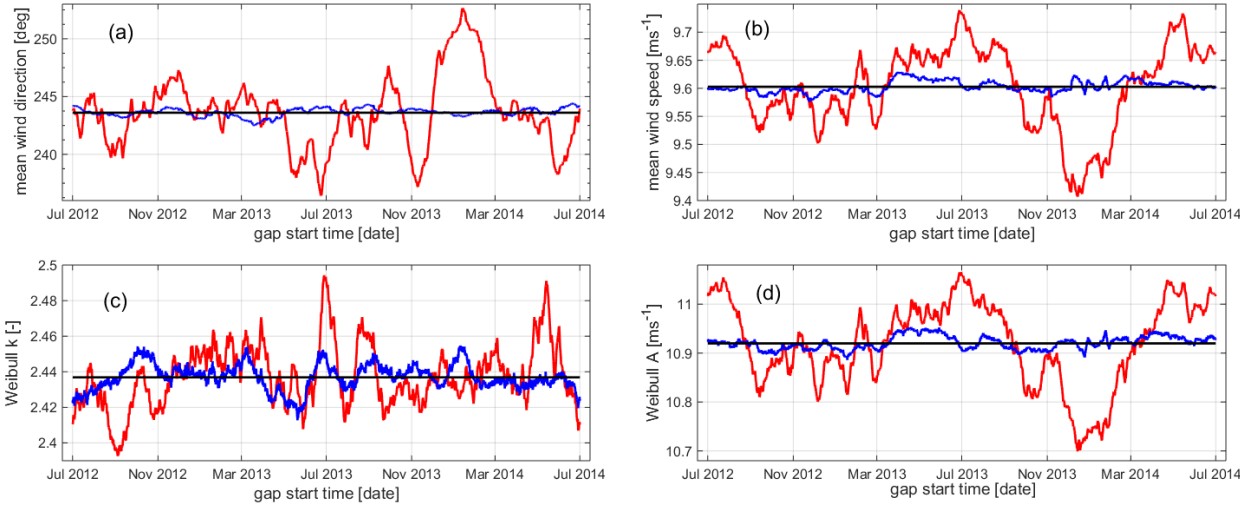

**Figure 6.** Variation of statistical measures – (a) mean wind direction and (b) mean wind speed, (c) Weibull shape and (d) scale parameter, $k$ and $A$ – depending on start date of artificial gap – here for gap length of 30 d, for incomplete time series in red and for filled time series in blue. Wind statistics for original time series in black as reference.

### 3.2.1 Long-term extrapolation

In a last step, the different time series are used as basis for a long-term extrapolation of the wind time series and statistics. Therefore, the measured time series with or without data gap are correlated with ERA5 reanalysis data that are available for a long-term period of 30 years in this case. The underlying MCP procedure is very similar to the one applied for the data gap

filling in 3.2. But this time the correlation period corresponds to the total measurement period of two years in our case (for the incomplete time series shortened by the gap length) and the prediction horizon to the complete 30 years for which the reference data is available. The measured time series has to be resampled to 1-hour data since the ERA5 data has no higher resolution. Otherwise, the same methods were used to derive and apply correction functions and offsets. As already pointed out above, we decided to use ERA5 data as long-term reference data (and not again WRF that is e.g. in the New European Wind Atlas also

available for a period of 30 years) because we believe this is a choice that still better corresponds to a typical case in a standard wind resource assessment application. The overall workflow followed in the study is summarized in Figure 7.

### 4 Results

In this section we present in some detail the results of the study expanding on the impact of gaps on the wind resources estimate with varying start dates (4.1) as well as of varying lengths (4.2), and the impact of the gaps on the long-term wind resource

estimate (4.3). Results are compared for the three considered sites Ijmuiden (Dutch North Sea), FINO2 (Baltic Sea) and FINO3 (German Bight).



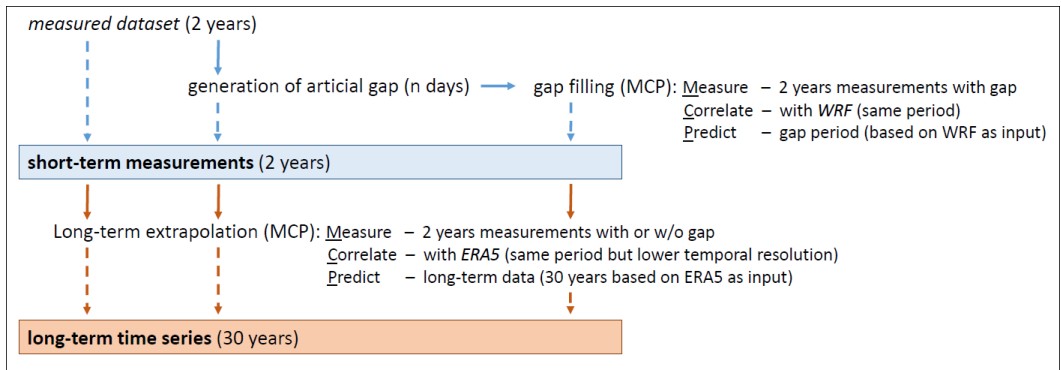

**Figure 7.** Workflow followed in our study including the MCP approaches for the gap filling and long-term extrapolation procedures, respectively.

## 4.1 Impact of gaps with varying start dates

Figure 8 shows how a 30-day gap impacts the four considered wind statistics (mean wind direction, mean wind speed, and the Weibull parameters $k$ and $A$) depending on the start date of the gap for all three studied sites. The results for FINO3, already presented in Figure 6, are shown in grey color, the results for Ijmuiden and FINO2 as dashed and dotted lines, respectively. Again, the applied gap filling (results in blue) reduces the deviations in the measures from the reference (in black) due to the existent gap (results in red) to a considerable degree. These reductions, quantified in terms of an RSME for the respective dataset of results, are summarized in Table 3. For mean wind direction, mean wind speed and Weibull scale parameter $A$ the

**Table 3.** RMSE derived for the four in this study considered statistics for gappy and gap-filled time series of wind speed and direction for the three investigated sites.

| Site | RMSE (mean dir – with gap) | RMSE (mean dir – gap filled) | RMSE (mean wsp – with gap) | RMSE (mean wsp – gap filled) |
|------|------|------|------|------|
| FINO3 | 3.1 deg | 0.3 deg | 0.07 ms$^{-1}$ | 0.01 ms$^{-1}$ |
| Ijmuiden | 2.3 deg | 0.3 deg | 0.09 ms$^{-1}$ | 0.01 ms$^{-1}$ |
| FINO2 | 2.2 deg | 0.4 deg | 0.07 ms$^{-1}$ | 0.01 ms$^{-1}$ |

| Site | RMSE (Weibull $k$ – with gap) | RMSE (Weibull $k$ – gap filled) | RMSE (Weibull $A$ – with gap) | RMSE (Weibull $A$ – gap filled) |
|------|------|------|------|------|
| FINO3 | 0.017 | 0.007 | 0.09 ms$^{-1}$ | 0.01 ms$^{-1}$ |
| Ijmuiden | 0.016 | 0.005 | 0.10 ms$^{-1}$ | 0.01 ms$^{-1}$ |
| FINO2 | 0.023 | 0.008 | 0.08 ms$^{-1}$ | 0.02 ms$^{-1}$ |

derived RMSE values, summarizing the deviations in the wind statistics due to the gaps with varying start dates, reduce up to a factor of ten. For the Weibull shape parameter $k$ this reduction is smaller (up to a factor of three) which is explained by the nature of this parameter. Overall, the reductions are similar for all three considered sites. Also the pattern of deviations in the wind statistics over the two-year course are pretty similar, except those for the Weibull $k$ parameter. Further to this, Figure 8 clearly shows how the wind statistics for the two sites FINO3 and FINO2, although having a very close mean wind speed for

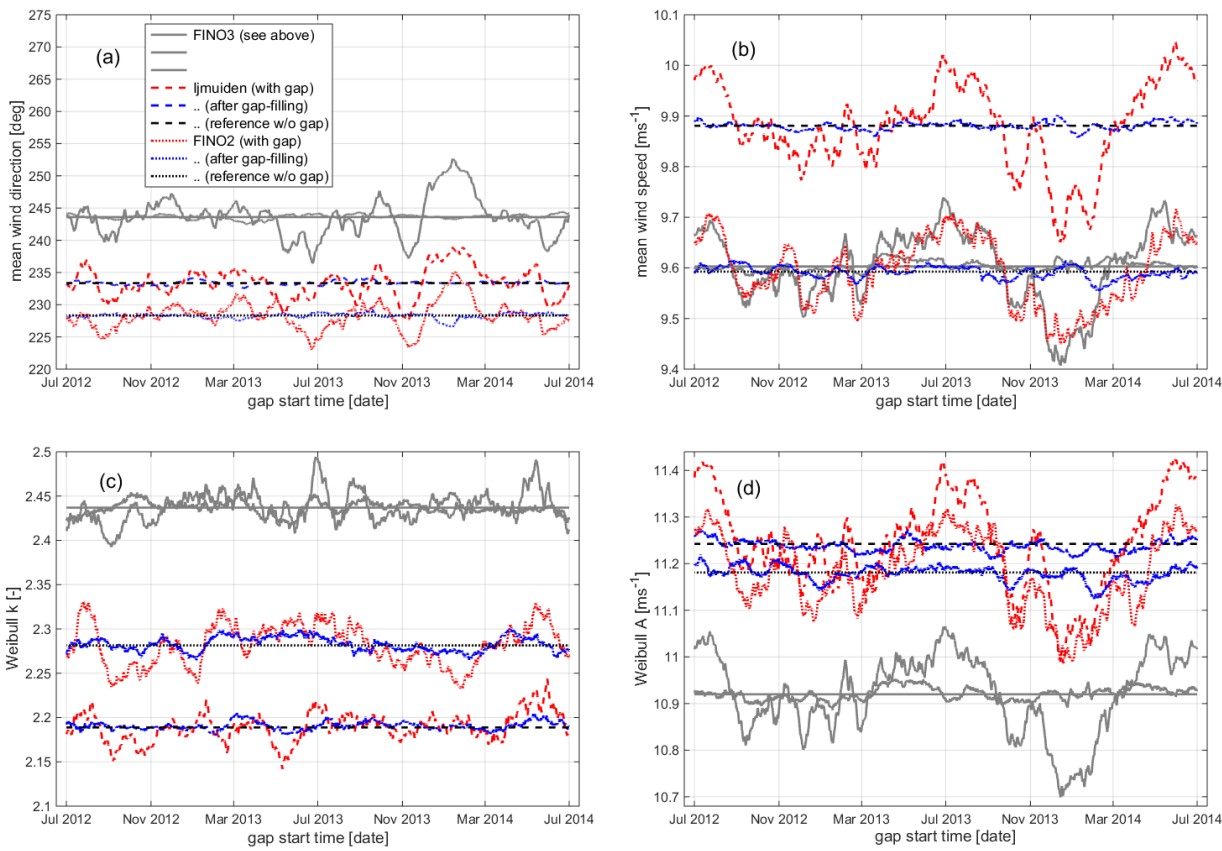

**Figure 8.** Variation of statistical measures – (a) mean wind direction and (b) mean wind speed, (c) Weibull shape and (d) scale parameter, $A$ and $k$, respectively – depending on start date of artificial gap, here for gap length of 30 d as in Figure 6. Results for FINO3 (already presented above) as solid grey lines, for Ijmuiden as dashed and for FINO2 as dotted lines (again for incomplete time series in red and for filled time series in blue, wind statistics for original time series in black).

the considered two-year period, differ with respect to their wind speed distributions and in particular the derived Weibull scale $A$ and shape $k$ parameters. The third site, Ijmuiden, in comparison, is characterized by both the highest mean wind speed and Weibull $A$ parameter and the lowest Weibull $k$ parameter.

## 4.2 Impact of gaps of varying lengths

5   In the next step, we have repeated this analysis for different gap lengths between 6 and 90 days. Figure 9 shows the derived RMSE values for the four considered wind statistics (in four separate plots) and the three sites (as solid, dashed and dotted curves) plotted against the gap length, again for ignored and filled gaps (in red and blue), respectively. In all considered cases, the impact of the gap (reflected by the derived RMSE) increases with gap length and is significantly reduced when applying



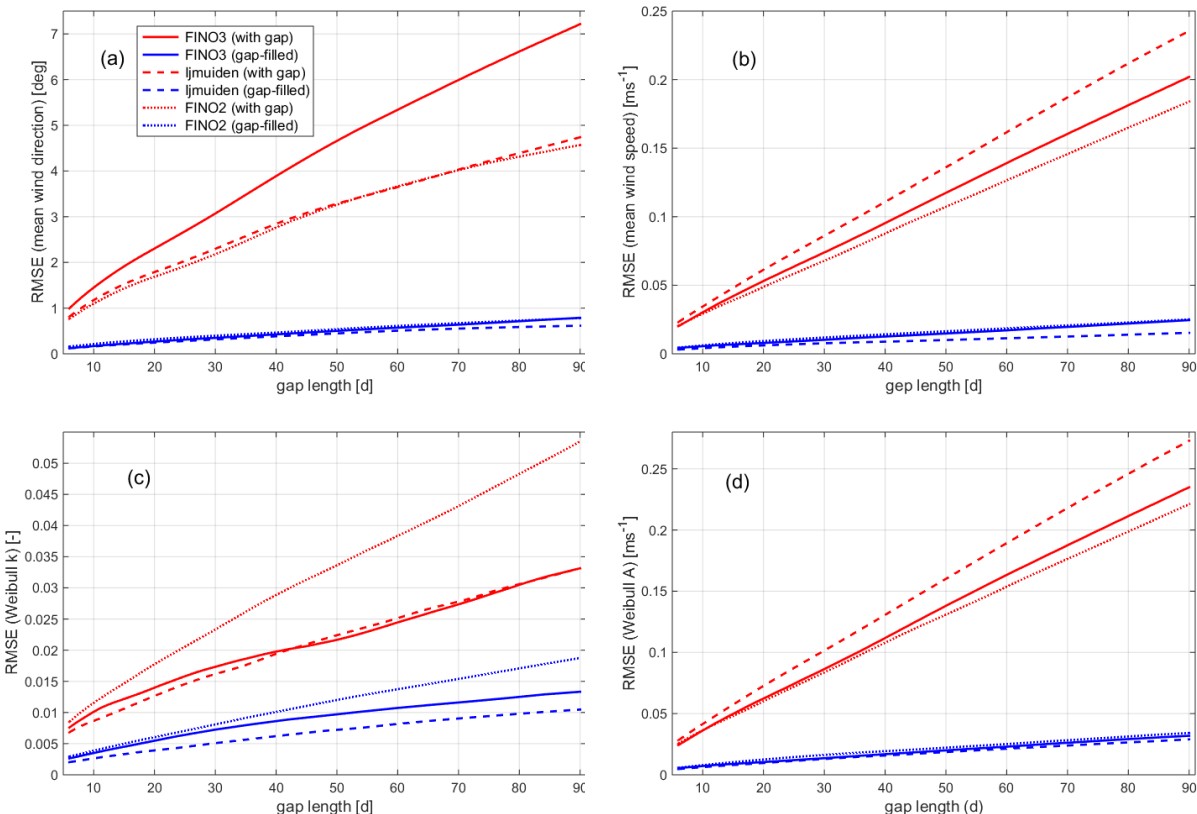

**Figure 9.** Dependency of impact of data gaps, quantified as RMSE of the four statistical measures – (a) mean wind direction and (b) mean wind speed, (c) Weibull shape and (d) scale parameter, $A$ and $k$, respectively – derived for gaps with systematically varying start dates, on the length of the data gaps. Results for FINO3 as solid lines, for Ijmuiden as dashed and for FINO2 as dotted lines (again for incomplete time series in red and for filled time series in blue).

the gap filling procedure. For the mean wind speed, just like for the Weibull $A$ parameter, the increase is more or less linear for the considered range up to 90 d, whereas is slightly flattens for the two other measures, mean wind direction and Weibull $k$.

Apart from these general agreements, the results for the three sites show also some deviations. For instance, the impact of the data gaps on the mean wind speed are largest, when these are ignored, for the Ijmuiden site but can be best compensated. This is shown by the smallest RMSE values, compared to those for the two other sites, after gap filling. This observation may be either explained by the performance of the used numerical model for the respective site or a statistical effect that relates to the level of observed wind speeds. (Remember Ijmuiden showed the highest measured mean wind speeds in the considered two-year period.)

The impact of (ignored) gaps in the wind direction time series is highest for the FINO3 dataset. This can be understood by looking again at the wind direction distributions in Figure 1: mean wind directions are more pronounced for the FINO2 and




Ijmuiden sites, whereas the distribution for FINO3 is characterized by a kind of site maximum for North-Westerly directions. A data gap may in this case remove data that correspond to a substantial part of one of the local maxima, having a larger impact on the overall distribution as for the case where the distribution has only one superior maximum range. After gap filling, the RMSE values are still larger for the FINO3 dataset than for the Ijmuiden data but the deviations are now much smaller. RMSE

values for FINO2 and FINO3 lie almost on top of each other.

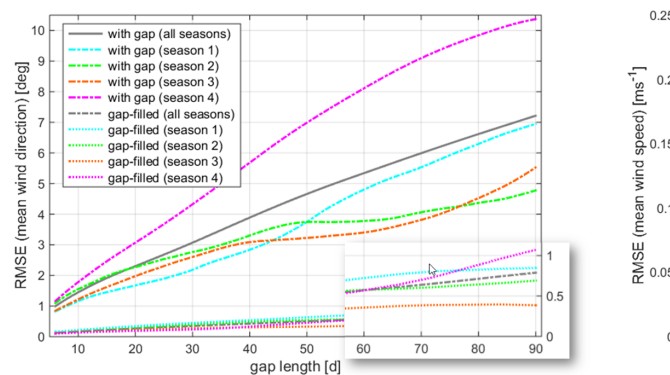
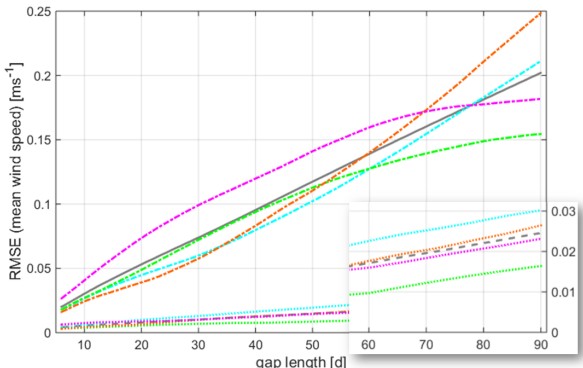

**Figure 10.** Dependency of impact of data gaps (quantified as RMSE as in Figure 9 but here only for mean wind direction, left, and mean wind speed, right) on the length of the data gaps and season (defined according to the gap start date). Results exemplarily for FINO3.

In a further step, we have studied how the impact of data gaps varies with the season in which the data gap occurs. For this, the "season" is defined by the start date of a gap – a gap starting in the months January to March is related to "season 1", one starting between April to June to "season 2" and so on. These seasons were selected with a shift of one month in comparison to the classical meteorological season definition of spring, summer, autumn and winter to consider the inertia of

the heating/cooling of the sea surface that mainly drives the yearly cycle of the atmospheric stability which vice versa has an impact on the wind distribution.

Figure 10 shows the results for FINO3 and the statistics mean wind speed and mean wind direction only, but they are more or less representative also for the other cases. The plots clarify that the impact of ignored and filled gaps significantly depends on the assigned season whereby also the performance of the gap filling shows a certain dependency, both not always going in

the same direction. Deviations are not only observed for the levels of derived RMSE values (i.e. how big is the impact) but also the shape of the curves (i.e. how does this change with gap length). This can be explained by the relation between gap length and the length of a season as defined above: a gap of a greater length is more likely to occur not just in the season it is assigned to. By this partly wrong assignment the seasonal effects are more mixed for the greater lengths.

### 4.3 Impact of gaps on long-term estimate

In a final step, we derive the impact of ignored and filled gaps in the measurement data on a long-term extrapolated mean wind speed. For this, we followed the procedure outlined in 3.2.1 and summarized in Figure 7. Figure 11 shows how the mean wind speed that is derived based on 30 years of ERA5 data and corrected according to the 2-year long measurements at the three





considered sites varies with the start date of a 30-d data gap that is cut into the measurement time series. Again the data gaps are either ignored (results in red) or filled by applying the introduced gap-filling procedure (results in blue). The following

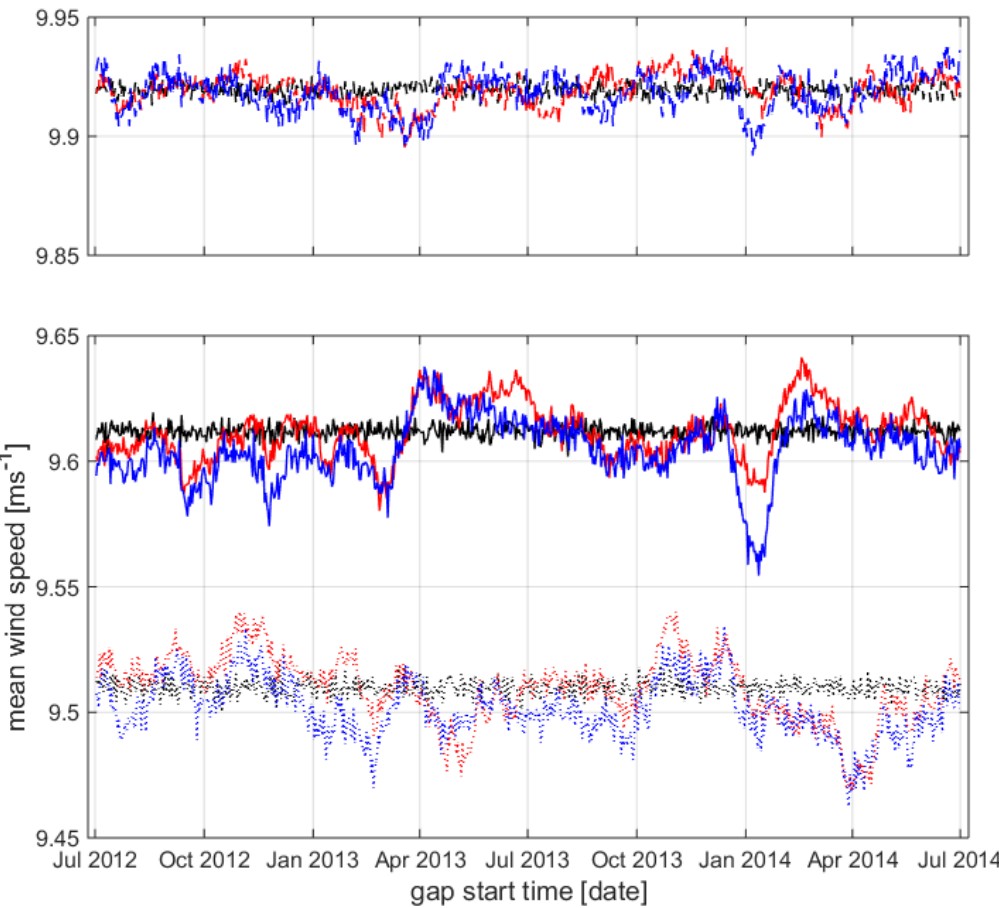

**Figure 11.** Variation of long-term corrected mean wind speed depending on start date of artificial 30-d gap in short-term measurements. Results for FINO3 as solid lines, for Ijmuiden as dashed and for FINO2 as dotted lines (again for incomplete short-term measurement time series in red and for filled time series in blue, for measurements without gap in black as reference).

conclusions can be drawn from Figure 11:

- The long-term corrected mean wind speed is significantly different from the mean values of the 2 years of measurements for the three considered sites. (Mean wind speed values of the used 30 years long ERA5 time series are 9.43 $\mathrm{ms}^{-1}$, 9.91 $\mathrm{ms}^{-1}$ and 9.26 $\mathrm{ms}^{-1}$ for FINO3, Ijmuiden and FINO2, respectively.)

- The impact of data gaps in the short-term measurements are visible in the long-term estimates but is rather small with an RMSE (for ignored gaps, red curves) of 0.011 $\mathrm{ms}^{-1}$ (FINO3), 0.007 $\mathrm{ms}^{-1}$ (Ijmuiden) and 0.014 $\mathrm{ms}^{-1}$ (FINO2), respectively.



- This impact of the data gaps in the short-term measurement on the long-term estimates is not really mitigated through the application of the gap filling procedure, corresponding RMSE values (for filled gaps, blue curves) are in the same range or even slightly larger with $0.015 \ \mathrm{ms}^{-1}$ (FINO3), $0.008 \ \mathrm{ms}^{-1}$ (Ijmuiden) and $0.014 \ \mathrm{ms}^{-1}$ (FINO2), respectively.

- Also the reference values (black curves) show some variability, that is due to the noise process as part of the MCP procedure applied for the long-term extrapolation. The RMSE values reflecting these variations are equal for all three sites with $0.003 \ \mathrm{ms}^{-1}$.

## 5   Discussion

Our study proposes a methodology that allows us to quantify the impact of data gaps in (measured) time series on wind statistics. With the three studied sites, we have considered three possible reference datasets, which could be referred to for

further sites where only incomplete time series are available but no suitable reference. The reference quantification can then be used to deduce an uncertainty associated to the inherent gaps, that e.g. could be related to the RMSE value derived for the variations in the wind statistics for different gap start dates for a fixed gap length. Alternatively, for a more conservative approach, the maximum deviations in the wind statistics observed in the reference study could be considered or, in case more details are available, the identified variations for a specific season.

For our study, we have only considered isolated single gaps in a measured time series. But the followed approach can be extended, in a straightforward way, to more complex scenarios including multiple gaps that may be more realistic or may correspond to a specific case of interest, respectively. We then would recommend the following procedure: the present scenario would first be generalised to an extent so that the available reference study case is sufficiently informative. If we want to evaluate the impact of a 20-day gap in February of a certain year, for instance, it may not be sufficient to study the impact of

such a gap in the reference data from another period only for the month of February. Instead the scenario may be broadened to a 20-day gap in the winter season. For making this decision, some background knowledge of the general wind climate at the studied sites is required, that can be gained e.g. from (numerical) long-term datasets. A similar approach is recommended for the consideration of multiple gaps, for which not only the lengths of the individual gaps need to be taken into account but also their distance in time and possible correlation effects.

With carrying out the study for three different offshore sites and showing the systematic similarities and some deviations between the results, we provided a basis for the selection of suitable reference sites and datasets. Again, some knowledge of the general wind climate at a site is required to evaluate whether a certain study site is suitable or not for the estimation of an uncertainty that is then used for the evaluation of the measurements from another site. In general, however, we believe that this transfer of observations is possible and suggest to use the available sites and datasets for this purpose. An extension of our

study to further sites, moreover, may help to better understand how the impact of data gaps on wind statistics may vary from site to site and to take such findings into account for an even more refined estimation of the associated uncertainties.

When looking at the mitigation of the impact of data gaps in the measured time series – explicitly, with the applied gap-filling procedure and the use of an MCP procedure in connection with wind data from a numerical model – we have only applied one





specific method but not further studied how the results may change with the application of other approaches. In this context, it was important for us to have a procedure that is straightforward and easy to apply for all three sites in exactly the same way. But we definitely also believe, that a refinement – e.g. by using more complex approaches and possibly also some fine-tuning for the individual sites – may show an optimised performance and with this less remaining impact of the data gaps on the wind

statistics after gap-filling. A specific gap-filling approach should be an integral part of the wind resource assessment process that is applied by a specific consultant for a specific site. At this it should also be pointed out that it is possibly not the optimal approach to apply the same type of MCP procedure for both the gap-filling and the long-term extrapolation step. Depending on whether the simulation of time series (i.e. for a point prediction or filling gaps) or the simulation of a wind distribution or wind statistics is of interest, so-called type I or type II MCP methods may be the better choice (Hanslian, 2017). In short, type

I MCP methods are designed to simulate time series whereas type II methods generate wind distributions.Whichever method is selected for the specific MCP task, this method should also be applied in the reference study to quantify the gap impact and estimate the associated uncertainty that is of high relevance for the (here: wind energy) application in any case.

In addition to this, it must also be kept in mind that the so quantified uncertainty is – when looking at the complete wind resource assessment process – not the only uncertainty that is associated to the long-term extrapolation. Another substantial

uncertainty component arises from the fact that the considered short-term period for which on-site measurements are available has only a limited representativeness for the long term. Some of this is compensated by the long-term extrapolation based on a "long" dataset itself but it needs to be considered that a derived correction function has always some dependency on the available correlation period. This dependency and related variations in the results of the estimated wind statistics constitute another part of the uncertainty associated to the long-term extrapolation, not yet taken into account, in a wind resource assessment.

**6  Conclusions**

As any field experiment, wind measurements that are typically carried out for site assessment studies are subject to data gaps due to e.g. failures of measurement devices or data loggers. In the harsh offshore wind climate, wind and wave conditions can lead to considerable time windows of inaccessibility of the measurement platform no matter if floating, e.g. buoy, or mast measurement. In our study we investigated the impact of these data gaps on typical statistical measures for wind energy siting

applications such as mean wind speed and direction and the Weibull shape and scale parameters. The study was performed for three offshore sites with meteorological mast measurements available between July 2012 and June 2014 in the southern North Sea (FINO3 and Ijmuiden) and the southern Baltic Sea (FINO2). We proposed a gap filling procedure that uses data from mesoscale meteorological modelling and studied the benefit of the gap filling in terms of the RMSE of the siting statistics. The study reports the following key results:

– A gap of 30 days in the dataset leads to an RMSE on the mean wind speed of up to about $0.1\,\mathrm{m\,s^{-1}}$ in the mean wind speed and the Weibull scale parameter $A$, an RMSE of about 0.02 on Weibull shape $k$ and $3\,\mathrm{deg}$ in the mean wind direction.



- The gap filling with mesoscale data can considerably reduce this impact up to a factor of three on the Weibull shape and a factor of ten on the three other investigated siting parameters mean wind speed, direction and Weibull scale parameter.

- The impact of the data gaps is monotonically and almost linearly increasing with the length of the data gap when considering the full year wind climate and so is the reduction of the impact of the gap filling. However, when looking at different seasons, the skill of the gap-filling differs.

- The key conclusions are similar for the three investigated sites, although the impact of gaps differ with the highest impact on the data from the FINO3 mast that is the only mast with a prominent impact of north-westerly winds and also the mast that is located furthest offshore.

- The impact of the gaps on the long-term estimate, expressed here in terms of a 30 year wind climatology is very small (around $0.01\,\mathrm{m\,s^{-1}}$ at all three sites) and cannot be substantially further reduced by the gap filling of the reference measurement dataset.

Our investigation focused on three European offshore sites in the North and Baltic Sea and could in future studies be evaluated for other offshore exploration areas with more different wind distributions in speed and direction. We intentionally focussed on three commonly used key wind energy siting statistics. With the tendency of a grid load based renumeration of wind power an investigation of the impact of data gaps on daily cycles might be interesting for future investigations.

*Code and data availability.* The mesoscale model data are available upon request, the mesoscale model itself is publicly available via github: https://github.com/wrf-model/WRF. The ERA5 data are available via the Copernicus Climate Data Store (CDS): https://cds.climate. copernicus.eu/cdsapp#!/dataset/reanalysis-era5-pressure-levels?tab=overview. The OSTIA data are available from the Copernicus Marine Environment Monitoring Service (CMEMS): https://resources.marine.copernicus.eu/?option=com_csw&task=results. The mast data are publicly available for scientific purposes via BSH and TNO.

*Author contributions.* JG performed the gap analysis and the implementation of filling and analysis procedures. MD prepared the measurement and reanalysis data and conducted the mesoscale model simulations for the gap filling. Both authors discussed the results and wrote and reviewed the manuscript.

*Competing interests.* The authors declare that they have no conflict of interest.

*Acknowledgements.* This research was partly carried out in the framework of the projects Digitale Windboje (ref. no. 03EE3024) and NEWA (ref. no. 0325832A) funded by the German Federal Ministry for Economic Affairs and Energy (BMWi) on the basis of a decision by the



German Bundestag with further financial support from NEWA ERA-NET Plus, topic FP7-ENERGY.2013.10.1.2, the latter only for NEWA. The simulations were performed at the HPC Cluster EDDY, located at the University of Oldenburg (Germany) and funded by BMWi (ref. no. 0324005). The study here was motivated by the results of two masters thesis projects: we acknowledge Bilke Engelbrecht and Christine Martens for their very valuable pre-works. We thank BSH for providing access to the FINO2 and FINO3 data, and TNO for the data of Ijmuiden met mast.



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
