# Peer review of "Understanding and mitigating the impact of data gaps on offshore wind resource estimates"

_Wind Energy Science, 2020_

## Referee Comment (RC1) · Anonymous Referee #1 · 16 Dec 2020

The paper presents a study of the impact, in terms of common wind-related error metrics, of gaps in measured time series of wind speed and direction, for wind resource estimates of offshore sites. The impact of gaps is studied both with and without gap-filling, using a measure-correlate-predict (MCP) technique in combination with numerical model data from a mesoscale model and a reanalysis dataset.

The paper is generally well written, sufficiently introduces the problem and the state-of-the-art, and presents the results in a clear way. However, some improvements should be made when describing the methods, to allow the readers to reproduce the results (see specific comments below).

[Figure]

The scope of the paper is somewhat narrow: three sites with fairly similar wind climates, one type of MCP method used, and just two years of measurements considered. Although this makes the paper easy to read and interpret, I believe it would be valuable to extend the scope, as the authors also hint at in the discussion, but perhaps that can be done in a subsequent paper. For instance, I would like to see periods longer than two years considered when seasonal errors due to gaps are investigated, as it can require more years to converge to "seasonal-average" representative results.

All in all, I found the paper interesting and valuable and would recommend accepting it with minor revisions.

**1 Specific comments**

- P1L6-7 - impact on what? the error? if so, what kind of error? as it stands, it's ambiguous. The abstract should speak for itself.

- P1L7-10 - again, it is unclear what "impact" means here, before it is clarified in P1L6-7

- P5L26 - You have chosen to investigate the impact on mean wind speed, direction, and Weibull A and k, but I think it would also be interesting to additionally show the impact on power, be it power density, or estimated production for reference turbines.

- P5L26 - How was the Weibull-fit made? e.g. maximum likelihood, or WAsP-like moments-based fit that preserves the wind power of the histogram? or something else?

- P6Fig3 - As I understand it, the Dantysk wind farm was constructed during 2014.

Did you consider whether the Fino3 measurements were disturbed by the emerging farm during days with flow from the east?

- P8 - Please also state what simulation duration and spin-up time was used with WRF

- P8 - It would be instructive to show the bias of the reference data compared to the measurements and how it varies with time, especially how it varies with season. If the reference data bias varies with season, which is often the case, it will probably be one of the dominating factors in explaining the MCP errors, particularly when the gap falls in one continuous period like here.

- P10L7 - Please be more specific in detailing the method. I assume you:

  – Use the modeled wind speed data as the predictor ($x$)
  – Use the measured wind speed data as the predictand ($y$)
  – group $x, y$ pairs (in time) by 0.5 m/s bins of $x$
  – calculate the mean and standard deviation of $y$ in each bin
  – make a piece-wise linear fit to the mean of $y$ and use that as the correction function

  Is this correct?

- P15 - As you mention, it may have been better to use the center point of the gaps as the reference time, as opposed to the period start. Especially when considering the season, since the central point better represents the time of the gap.

- P16 - Can you offer a deeper explanation for why the gap-filling before long-term extrapolation leads to equally large errors as not gap-filling first? it seems counter-intuitive

- P17L5-6 - How is the RMSE $= 0.003$ for the black curves calculated here? in other words, what is considered the "target" result? the mean?

**2 Technical corrections**

- P2L10 - "threshold of amount of lengths of data gaps", perhaps just "threshold of lengths of data gaps"?

- P3L13-14 - "And with Section 6 we conclude our contribution." to me it sounds like you put the emphasis on ending the paper, rather than concluding on the results of the study. I would suggest rephrasing it.

- P3L23 - "next" → "nearest"?

- P4Fig1 and P8Fig4 - Please make sure you adhere to the guidelines related to copyrights and distribution licenses for the background maps. See the author guidelines.

- P5Fig5 and P15Fig10 - Please add panel labeling, e.g. (a), (b), etc.

- P7L23 - Please spell out "grid points (GP)" the first time it's used

- P7L23 - The "x" symbol seems like a regular x rather than a "times" symbol, e.g. like LaTeX's $\times$ made using: `$\times$`

- P9L7 - Note that colons are recommended between hours, minutes, and seconds. Example from the guidelines: 25 July 2007 (dd month yyyy), 15:17:02 (hh:mm:ss)

- P10L8-10 - The explanation seems more convoluted than it needs to be. Why not state that you used the piece-wise linear fit as the correction function?

- P15L14 - "both" → "but"?

- P18L6 - Incomplete sentence. "At this it should also be"...

- P18L13 - Should "so" be removed here?

- Everywhere: please use the abbreviation "Fig." in running text, as per the Author guidelines.

- Everywhere: I would suggest adding a space, e.g. `$\,$` with LaTeX, between units, e.g. m s$^{-1}$ rather than ms$^{-1}$

---

## Referee Comment (RC2) · Anonymous Referee #2 · 10 Jan 2021

The paper is motivated by the impact of the measurement gaps due to device failures. The paper could easily be extended to also the onshore situations, and it would be interesting and relevant to see how successful the gap-filling can be when the measurement data characteristics are similar to that from the typical remote sensing campaigns in the wind resource assessment practice. It would also be very informative if the impact of the several shorter gaps would be calculated, and compared with the impact of a long gap with equal duration as the combined shorter gaps (it can happen on a met mast, in winter time, that several shorter gaps occur due to e.g. icing).

The final results (Figure 11) are rather surprising: the gap-filling procedure does not

help long-term extrapolation, and does not harm it at best. Thus the conclusion must be that gap-filling is not necessary for resource assessment (for the 30-day gap though), opposite to the authors' claim in the Discussion chapter, P18L5. The main contribution of this paper is in demonstrating how succesful gap-filling can be in the reconstruction of a wind time-series with gaps.

The paper is clearly written and otherwise acceptable with a minor revision.

Specific comments

*Abstract*

- P1L6-7: sentence unclear "mitigation of the gaps' impact by a factor of ten ..." Introduction,

- P1L12: wind resource also is often re-assessed based on the WTG data, and sometimes a met-mast, all of which also may have gaps.

- P1L15-16: would be interesting to quantify the impact of the uncertainty in terms of the wind project value, e.g. a 1% uncertainty in the wind speed that results in 2% or more uncertainty of the wind farm production which is almost directly related to the earning and compared to an expected return of investment of e.g. 8%.

*Data basis*

- P4L3-11: Filling gaps with data from below scaled by the ratio of mean wind speeds is unnecessarily rough. Luckily the wind speed gaps in the raw measuremetns at the considered heights of 92 m are very short so this probably does not have a significant impact to the outcome of the whole wind speed analysis. Nevertheless, if possible, it would be necessary to apply a more refined method (e.g. using the extrapolated wind profile from two data points below on a 10-minute basis).

- Table 1: The wind distribution parameters A and k are presented, but what are their confidence intervals (Weibull fitting is not perfect because the wind is not entirely

Weibull distributed)? Further analysis in the paper should then take these confidence intervals into account when the impact of the gaps, and of their mitigation, is presented and discussed. (e.g. P10L30, the error of the k is reduced from 0.017 to 0.007, but what if the uncertainty of the computed k is e.g. 0.1 in the first place?).

- P7L3: Please use the common terminology for NWP and climate modelling: "longitudinal" –> zonal, "lateral" –> meridional

*Applied procedures*

- P9L6-7: "start state" or perhaps "start date"?

- P10L14-15: while I agree that it would not add much to the study to use a sector-wise approach in the linear correction, it is probably not completely true that 50 km is a sufficient distance (from the coast) that such an approach would not be necessary. Please delete this claim.

- P10L16-20: Please show the magnitude of the noise applied. Applying a random noise in such a way poses a risk that the resulting time-series loses its physical consistency.

- Figure 6 (and associated analysis): is there any dependency of the gap-filling success on the WRF errors (as compared with the non-gapped met mast data)? I.e. there is a certain match between the gap impact itself (red line) and the error of the corrected series (blue), but what makes this correlation not perfect? Model error, or another (stochastic) effect?

- P11, section 3.2.1.: please clarify if WRF is also used for gap-filling when ERA5 is used for the long-term extrapolation?

- P11L10: Using ERA5 directly for MCP is authors' choice and should be stated like this. Typical case is to use downscaled e.g. ERA5 (Windpro/ConWX, Vortex, ...).

*Results*

- Table 3: again, these results should be accompanied by the underlying uncertainty in the calculation of the statistical parameters.

*Discussion*

- P18L5: why do you say that gap-filling should be an integral part of wind resource assessment?

- Figure 11 shows that gap-filling or not has no impact on the long-term wind assessment.

---

## Author Response (AR1)

**Response to Review**

**Reviewer 1**

The paper presents a study of the impact, in terms of common wind-related error metrics, of gaps in measured time series of wind speed and direction, for wind resource estimates of offshore sites. The impact of gaps is studied both with and without gap-filling, using a measure-correlate-predict (MCP) technique in combination with numerical model data from a mesoscale model and a reanalysis dataset.

The paper is generally well written, sufficiently introduces the problem and the state-of-the-art, and presents the results in a clear way. However, some improvements should be made when describing the methods, to allow the readers to reproduce the results (see specific comments below). The scope of the paper is somewhat narrow: three sites with fairly similar wind climates, one type of MCP method used, and just two years of measurements considered. Although this makes the paper easy to read and interpret, I believe it would be valuable to extend the scope, as the authors also hint at in the discussion, but perhaps that can be done in a subsequent paper. For instance, I would like to see periods longer than two years considered when seasonal errors due to gaps are investigated, as it can require more years to converge to "seasonal-average" representative results.

All in all, I found the paper interesting and valuable and would recommend accepting it with minor revisions.

We thank Reviewer 1 for the careful evaluation and believe the suggested changes, implemented by us within the revision process, have improved the manuscript considerably. We also agree that the scope of the paper is rather narrow and could be extended in a subsequent or follow-up paper. In fact, our main focus has been to present our approach in an as reduced as possible form so that the associated procedure can be applied in as many cases as possible (in later studies). We believe that the extensions are straightforward to implement.
Please find our detailed answers to all the comments separately below.

**Specific Comments:**

[**R1SC1**]: P1L6-7 - impact on what? the error? if so, what kind of error? as it stands, it's ambiguous. The abstract should speak for itself
We assume the reviewer referred to the wrong line here (should be L4). We have added: "... in terms of a bias in the estimation of siting parameters".

[**R1SC2**]: P1L7-10 - again, it is unclear what "impact" means here, before it is clarified in P1L6-7
We agree and have added: "i.e. a reduction of the observed biases"

[**R1SC3**]: P5L26 - You have chosen to investigate the impact on mean wind speed, direction, and Weibull A and k, but I think it would also be interesting to additionally show the impact on power, be it power density, or estimated production for reference turbines.
Yes, we agree. We have added a corresponding paragraph to the Discussion section and reproduced Fig. 6 for the wind power density.

[**R1SC4**]:P5L26 - How was the Weibull-fit made? e.g. maximum likelihood, or WAsP-like moments-based fit that preserves the wind power of the histogram? or something else?
Thanks for asking for this clarification. We have added: "The fitting procedure is implemented as a nonlinear least-squares regression considering the complete wind speed range."

[**R1SC5**]: P6Fig3 - As I understand it, the Dantysk wind farm was constructed during 2014. Did you consider whether the Fino3 measurements were disturbed by the emerging farm during days

with flow from the east?

According to [1] the wind farm fed its first kilo Watt hour in December 2014. The first turbine was installed in April 2014 while the operation of single turbines did not start before autumn according to [2]. The time windows we have investigated covers data until June 30th, 2014. We have added a sentence to clarify this to the introduction of the FINO3 mast: "Less than a kilometer west of the FINO3 the wind farm DanTysk was constructed between February 2013 - April 2015. The erection of turbines did not start before April 2014 and operation not before December 2014. So the wind statistics of FINO3 should not be impacted by wakes of DanTysk."

[1] https://powerplants.vattenfall.com/dantysk
[2] https://www.offshore-windindustrie.de/news/nachrichten/artikel-26061-vattenfall-meldet-erste-turbine-im-offshore-windpark-dantysk-und-testet-neue-installations-technik.

**[R1SC6]:** P8 - Please also state what simulation duration and spin-up time was used with WRF
Simulation duration was 10 days (240 hours) plus 24 hours additonal spin-up time. We have added this information to Table 2 in the revised version of the manuscript.

**[R1SC7]:** P5L26 P8 - It would be instructive to show the bias of the reference data compared to the measurements and how it varies with time, especially how it varies with season. If the reference data bias varies with season, which is often the case, it will probably be one of the dominating factors in explaining the MCP errors, particularly when the gap falls in one continuous period like here.
We have done this analysis but not included it in the paper: we could not find a significant bias between the numerical data and measurements. Monthly mean biases (for FINO3 as an example here) vary between 0.38 and 0.50 $\mathrm{ms}^{-2}$ with the largest value for August and the smallest for April.

**[R1SC8]:** P10L7 - Please be more specific in detailing the method. I assume you:
– Use the modeled wind speed data as the predictor (x)
– Use the measured wind speed data as the predict and (y)
– group x; y pairs (in time) by 0.5 m/s bins of x
– calculate the mean and standard deviation of y in each bin
– make a piece-wise linear fit to the mean of y and use that as the correction function
Is this correct?
Thanks for pointing out that this method description was obviously not clear enough. We have rewritten this part as follows: "For the wind speed data, we – first – bin the wind speeds every $0.5\,\mathrm{m\,s}^{-1}$ based on the modeled data and calculate the average measured values in every bin. Second, we fit two linear functions for the wind speed ranges $[0, 5)$ and $[5, 20]$ $\mathrm{m\,s}^{-1}$. The resulting coefficients of the linear fits are then applied to correct the respective modelled wind speed and account this way for the systematic error between measured and modelled data."

**[R1SC9]:** P15 - As you mention, it may have been better to use the center point of the gaps as the reference time, as opposed to the period start. Especially when considering the season, since the central point better represents the time of the gap.
We do not think that this alternative approach may have been better – the situation that a (long) data gap takes place in two seasons would remain. Actually we think that the chosen approach is better because it links the gap to the failure reason (for which the starting date may be more meaningful than a centre point).

**[R1SC10]:** P16 - Can you offer a deeper explanation for why the gap-filling before longterm extrapolation leads to equally large errors as not gap-filling first? it seems counter-intuitive
We added this explanation to the discussion in the revised version: "The fact that we have not optimised the MCP methods for our applications may also be the reason for the initially counter-intuitive observation that the gap filling procedure, applied to the short-term measurements, has no positive effect on the long-term extrapolated results (cf. Sect. 4). Another reason is the relatively short gap of only one month which is still within the availability of $> 90\%$ accepted by MEASNET."

**[R1SC11]:** P17L5-6 - How is the RMSE = 0:003 for the black curves calculated here? In other words, what is considered the "target" result? the mean?
Thanks for asking for this clarification. We have added: "where the mean value is considered as reference".

**Technical Corrections:**

**[RC1TC1]:** P2L10 - "threshold of amount of lengths of data gaps", perhaps just "threshold of lenghts of data gaps"?
Thank you. We have modified the text to "Up to a certain threshold of frequency and length of data gaps".

**[RC1TC2]:** P3L13-14 - "And with Section 6 we conclude our contribution." to me it sounds like you put the emphasis on ending the paper, rather than concluding on the results of the study. I would suggest rephrasing it.
Thank you. We have modified the text to "And, finally, in Section 6 we summarize the main conclusions of our study."

**[RC1TC3]:** P3L23 - "next" $\rightarrow$ "nearest"?
Changed accordingly. Thanks.

**[RC1TC4]:** P4Fig1 and P8Fig4 - Please make sure you adhere to the guidelines related to copyrights and distribution licenses for the background maps. See the author guidelines.
Thank you. We have added the reference for the data source (GSHHS) to both captions

**[RC1TC5]:** P5Fig5 and P15Fig10 - Please add panel labeling, e.g. (a), (b), etc.
Thanks for the advice. For Fig. 5 we have not added panel labelings since the three plots result in one illustration. For Fig. 10 we have added the labels.

**[RC1TC6]:** P7L23 - Please spell out "grid points (GP)" the first time it's used
Changed accordingly. Thanks.

**[RC1TC7]:** P7L23 - The "x" symbol seems like a regular x rather than a "times" symbol, e.g. like LATEX's made using: $\times$
Changed accordingly. Thanks.

**[RC1TC8]:** "P9L7 - Note that colons are recommended between hours, minutes, and seconds. Example from the guidelines: 25 July 2007 (dd month yyyy), 15:17:02 (hh:mm:ss)"
Changed accordingly. Thanks.

**[RC1TC9]:** P10L8-10 - The explanation seems more convoluted than it needs to be. Why not state that you used the piece-wise linear fit as the correction function?
This is the same position as RC1SC8. Please have a look at the clarification we suggested there.

**[RC1TC10]:** P15L14 - "both" $\rightarrow$ "but"?
Changed accordingly. Thanks.

**[RC1TC11]:** P18L6 - Incomplete sentence. "At this it should also be"...
Changed accordingly. Thanks.
We have removed the "At this" to make it a meaningful sentence. Thanks

[**RC1TC12**]: P18L13 - Should "so" be removed here?
Changed accordingly. Thanks.

[**RC1TC13**]: Everywhere: please use the abbreviation "Fig." in running text, as per the Author guidelines
Changed accordingly. Thanks.

[**RC1TC14**]: Everywhere: I would suggest adding a space, e.g. with LATEX, between units, e.g. m s-1 rather than ms-1
Changed accordingly everywhere. Thanks.

**Reviewer 2**

The paper is motivated by the impact of the measurement gaps due to device failures. The paper could easily be extended to also the onshore situations, and it would be interesting and relevant to see how successful the gap-filling can be when the measurement data characteristics are similar to that from the typical remote sensing campaigns in the wind resource assessment practice. It would also be very informative if the impact of the several shorter gaps would be calculated, and compared with the impact of a long gap with equal duration as the combined shorter gaps (it can happen on a met mast, in winter time, that several shorter gaps occur due to e.g. icing). The final results (Figure 11) are rather surprising: the gap-filling procedure does not help long-term extrapolation, and does not harm it at best. Thus the conclusion must be that gap-filling is not necessary for resource assessment (for the 30-day gap though), opposite to the authors' claim in the Discussion chapter, P18L5. The main contribution of this paper is in demonstrating how succesful gap-filling can be in the reconstruction of a wind time-series with gaps.

The paper is clearly written and otherwise acceptable with a minor revision.

We thank Reviewer 2 for the positive words about our manuscript and the thorough evaluation. We address every point raised by the reviewer separately below.

**Specific Comments:**

[**RC2SC1**]: P1L6-7: sentence unclear "mitigation of the gaps' impact by a factor of ten ..."
Thank you. In the revised version of the manuscript, we have changed this sentence to "We find a mitigation of the data gaps' impact, i.e. a reduction of the observed biases, by a factor of ..."

[**RC2SC2**]: P1L12: wind resource also is often re-assessed based on the WTG and sometimes a met-mast, all of which also may have gaps.
We have added the following sentence to account for the comment of the reviewer: "During the lifetime of a wind farm re-assessments are also typically done that can be based on wind turbine or further wind measurement data."

[**RC2SC3**]: P1L15-16: would be interesting to quantify the impact of the uncertainty in terms of the wind project value, e.g. a 1% uncertainty in the wind speed that results in 2% or more uncertainty of the wind farm production which is almost directly related to the earning and compared to an expected return of investment of e.g. 8%.
Good point. We have added the sentences "Consequently, uncertainties and a possible bias in the wind resource estimate propagate up to the financing of a wind project with the percentage uncertainty value increasing from uncertainty in wind speed to uncertainty in wind farm production to uncertainty in the expected return on investment. Thus to reduce these uncertainties starting

from the wind measurements is of high interest and relevance." Sorry we could not find a suitable reference for explicit numbers.

**[RC2SC4]:** P4L3-11: Filling gaps with data from below scaled by the ratio of mean wind speeds is unnecessarily rough. Luckily the wind speed gaps in the raw measuremetns at the considered heights of 92 m are very short so this probably does not have a significant impact to the outcome of the whole wind speed analysis. Nevertheless, if possible, it would be necessary to apply a more refined method (e.g. using the extrapolated wind profile from two data points below on a 10-minute basis).
As shown in Figure 2 the data sets were very complete even before the gap filling applied here. In case of FINO3 and Ijmuiden the filling was less than 0.5 % of the values. The measurement period selected was also based on consideration of the availability of the mast data. We didn't aim for having a precise filling of these small gaps here but rather tried to generate a complete time series for the methodology in the later steps.

**[RC2SC5]:** Table 1: The wind distribution parameters A and k are presented, but what are their confidence intervals (Weibull fitting is not perfect because the wind is not entirely Weibull distributed)? Further analysis in the paper should then take these confidence intervals into account when the impact of the gaps, and of their mitigation, is presented and discussed. (e.g. P10L30, the error of the k is reduced from 0.017 to 0.007, but what if the uncertainty of the computed k is e.g. 0.1 in the first place?).
We agree that the confidence intervals should be considered when integrating such an analysis in a WRA study (in particular, when assessing the final uncertainties of the wind resource estimates) but for our study it is out of scope. For offshore conditions, the Weibull distribution is a pretty good approximation.

**[RC2SC6]:** P7L3: Please use the common terminology for NWP and climate modelling: "longitudinal" $\rightarrow$ zonal, "lateral" $\rightarrow$ meridional
Changed Accordingly. Thanks.

**[RC2SC7]:** P9L6-7: "start state" or perhaps "start date"?
It should be "start" only. We removed the "state". Thanks.

**[RC2SC8]:** P10L14-15: while I agree that it would not add much to the study to use a sectorwise approach in the linear correction, it is probably not completely true that 50 km is a sufficient distance (from the coast) that such an approach would not be necessary. Please delete this claim.
Deleted as suggested by the reviewer. Thank you.

**[RC2SC9]:** Please show the magnitude of the noise applied. Applying a random noise in such a way poses a risk that the resulting time-series loses its physical consistency.
Thanks for pointing this out. For clarification we have added the sentence "The noise factor ensures that the generated time series does not lose its physical consistency." after "... a noise factor is derived as standard deviation of the data per bin, and combined with a white-noise process in the prediction step." to the revised version of the manuscript.

**[RC2SC10]:** Figure 6 (and associated analysis): is there any dependency of the gap-filling success on the WRF errors (as compared with the non-gapped met mast data)? I.e. there is a certain match between the gap impact itself (red line) and the error of the corrected series (blue), but what makes this correlation not perfect? Model error, or another (stochastic) effect?
We have investigated the bias between the original numerical (WRF) data and the observations and could not identify any significant dependency (cf. RC1SC7). Instead, we believe the varying "success" of the gap-filling procedure depends on how representative the correlation period is for the gap, i.e. the period that needs to be predicted within the MCP framework. If the gap corresponds to a significant part of a season, this part will not be well represented in the data

basis used for correction. A refined MCP approach may balance this deficit at least to some extent.

**[RC2SC11]:** P11, section 3.2.1.: please clarify if WRF is also used for gap-filling when ERA5 is used for the long-term extrapolation?
Thanks for pointing this out. We have added the half-sentence ".., while the gap-filling is still based on the mesoscale model data as before."

**[RC2SC12]:** P11L10: Using ERA5 directly for MCP is authors' choice and should be stated like this. Typical case is to use downscaled e.g. ERA5 (Windpro/ConWX, Vortex, ...).
We agree that at onshore sites, downscaled data are more commonly used. For offshore sites, newer studies showed that ERA5 data already provide very good results, see e.g.

Dörenkämper, M., Olsen, B. T., Witha, B., Hahmann, A. N., Davis, N. N., Barcons, J., Ezber, Y., García-Bustamante, E., González-Rouco, J. F., Navarro, J., Sastre-Marugán, M., Sīle, T., Trei, W., Žagar, M., Badger, J., Gottschall, J., Sanz Rodrigo, J., and Mann, J.: The Making of the New European Wind Atlas – Part 2: Production and evaluation, Geosci. Model Dev., 13, 5079–5102, https://doi.org/10.5194/gmd-13-5079-2020, 2020.

**[RC2SC13]:** Table 3: again, these results should be accompanied by the underlying uncertainty in the calculation of the statistical parameters.
Thanks. We have added the sentence: "Uncertainties or standard errors in the estimation of the parameters are not further considered here and in the following as they are small compared to the reduction of the gap impact which is the focus of this study." at the end of 3.2.

**[RC2SC14]:** P18L5: why do you say that gap-filling should be an integral part of wind resource assessment?
According to MEASNET Guideline "Evaluation of Site Specific Wind Conditions" data gaps should in general be filled, although the recommendations therein focus on filling those with data from other heights or different sensors and are thus more related to the classical onshore approach of met mast measurements. We show in our study, that the gap filling does not improve the long-term assessment. However, our data gaps were rather short, which could change for gaps that cover e.g. periods of a whole season or when not only the long-term statistics are relevant but average time series, e.g. diurnal wind speed changes are of interest. Thus, we suggest to modify the sentence the reviewer referred to: "We believe that a specific gap-filling approach should be an integral part of the wind resource assessment process that is applied by a specific consultant for a specific site as it improves the wind statistics of the measured period and can potentially also reduce the uncertainty of the long-term assessment."

**[RC2SC15]:** Figure 11 shows that gap-filling or not has no impact on the long-term wind assessment.
This is correct. After a comment of Reviewer 1 (see [R1SC10], we put more emphasis on this (quite central) finding.